# Enabling hyperparameter optimization in sequential autoencoders for spiking neural data

**Mohammad Reza Keshtkaran**
Coulter Dept. of Biomedical Engineering
Emory University and Georgia Tech
Atlanta, GA 30322
mkeshtk@emory.edu

**Chethan Pandarinath**
Coulter Dept. of Biomedical Engineering
Dept of Neurosurgery
Emory University and Georgia Tech
Atlanta, GA 30322
chethan@gatech.edu

## Abstract

Continuing advances in neural interfaces have enabled simultaneous monitoring of spiking activity from hundreds to thousands of neurons. To interpret these large-scale data, several methods have been proposed to infer latent dynamic structure from high-dimensional datasets. One recent line of work uses recurrent neural networks in a sequential autoencoder (SAE) framework to uncover dynamics. SAEs are an appealing option for modeling nonlinear dynamical systems, and enable a precise link between neural activity and behavior on a single-trial basis. However, the very large parameter count and complexity of SAEs relative to other models has caused concern that SAEs may only perform well on very large training sets. We hypothesized that with a method to systematically optimize hyperparameters (HPs), SAEs might perform well even in cases of limited training data. Such a breakthrough would greatly extend their applicability. However, we find that SAEs applied to spiking neural data are prone to a particular form of overfitting that cannot be detected using standard validation metrics, which prevents standard HP searches. We develop and test two potential solutions: an alternate validation method ("sample validation") and a novel regularization method ("coordinated dropout"). These innovations prevent overfitting quite effectively, and allow us to test whether SAEs can achieve good performance on limited data through large-scale HP optimization. When applied to data from motor cortex recorded while monkeys made reaches in various directions, large-scale HP optimization allowed SAEs to better maintain performance for small dataset sizes. Our results should greatly extend the applicability of SAEs in extracting latent dynamics from sparse, multidimensional data, such as neural population spiking activity.

## 1 Introduction

Over the past decade, our ability to monitor the simultaneous spiking activity of large populations of neurons has increased exponentially, promising new avenues for understanding the brain. These capabilities have motivated the development and application of numerous methods for uncovering dynamical structure underlying neural population spiking acqtivity, such as linear or switched linear dynamical systems [1, 2, 3, 4, 5], Gaussian processes [6, 7, 8, 9], and nonlinear dynamical systems [10, 11, 12, 13]. With this rich space of models, several factors influence which model is most appropriate for any given application, such as whether the data can be well-modeled as an autonomous dynamical system, whether interpretability of the dynamics is desirable, whether it is important to link neural activity to behavioral or task variables, and simply the amount of data available.

We previously developed a method known as Latent Factor Analysis via Dynamical Systems (LFADS), which used recurrent neural networks in a modified sequential autoencoder (SAE) configuration to uncover estimates of latent, nonlinear dynamical structure from neural population spiking activity [10, 12]. LFADS inferred latent states that were predictive of animals' behaviors on single trials, inferred perturbations to dynamics that correlated with behavioral choices, linked spiking activity to oscillations present in local field potentials, and combined data from non-overlapping recording sessions that spanned months to improve inference of underlying dynamics. These features may be useful for studying a wide range of questions in neuroscience. However, SAEs have very large parameter counts (tens to hundreds of thousands of parameters), and this complexity relative to other models has raised concerns that SAEs (and other neural network-based approaches) may only perform well on very large training sets [8]. We hypothesized that properly adjusting model hyperparameters (HPs) might increase the performance of SAEs in cases of limited data, which would greatly extend their applicability. However, when we attempted to test the adjustment of SAE HPs beyond their previous hand-tuned settings, we found that SAEs are susceptible to overfitting on spiking data. Importantly, this overfitting could not be detected through standard validation metrics. Conceptually, one knows that the best possible autoencoder is a trivial identity transformation of the data, and complex models with enough capacity can converge to this solution. Without knowing the dimensionality of the latent dynamic structure *a priori*, it is unclear how to constrain the autoencoder to avoid overfitting while still providing the capacity to best fit the data. Thus, while it may be possible to manually tune HPs and achieve better SAE performance (e.g., by visual inspection of the results), building a framework to optimize HPs in a principled fashion and without manual intervention remains a key challenge.

This paper is organized as follows: Section 2 demonstrates the tendency of SAEs to overfit on spiking data; Section 3 proposes two potential solutions to this problem and characterizes their performance on simulated datasets; Section 4 demonstrates the effectiveness of these solutions through large-scale HP optimization in applications to motor cortical population spiking activity.

## 2 Sensitivity of SAEs to overfitting on spiking data

### 2.1 The SAE architecture

We examine the LFADS architecture detailed in [10, 12]. The basic model is an instantiation of a variational autoencoder (VAE) [14, 15] extended to sequences, as in [16, 17, 18]. Briefly, an encoder RNN takes as input a data sequence $\mathbf{x}_t$, and produces as output a conditional distribution over a latent code $\mathbf{z}$, $Q(\mathbf{z}|\mathbf{x}_t)$. In the VAE framework, an uninformative prior $P(\mathbf{z})$ on this latent code serves as a regularizer, and divergence from the prior is discouraged via a training penalty that scales with $D_{KL}(Q(\mathbf{z}|\mathbf{x}_t)\|P(\mathbf{z}))$. A data sample $\hat{\mathbf{z}}$ is then drawn from $Q(\mathbf{z}|\mathbf{x}_t)$, which sets the initial state of a decoder RNN. This RNN attempts to create a reconstruction $\hat{\mathbf{r}}_t$ of the original data via a low-dimensional set of factors $\hat{\mathbf{f}}_t$. Specifically, the data $\mathbf{x}_t$ are assumed to be samples from an inhomogenous Poisson process with underlying rates $\hat{\mathbf{r}}_t$. This basic sequential autoencoder is appropriate for neural data that is well-modeled as an autonomous dynamical system.

Our previous work also demonstrated modifications of the SAE architecture for modeling input-driven dynamical systems (Figure 1(a), also detailed in [10, 12]). In this case, an additional controller RNN compares an encoding of the observed data with the output of the decoder RNN, and attempts to inject a time-varying input $\mathbf{u}_t$ into the decoder to account for data that cannot be modeled by the decoder's autonomous dynamics alone. (As with the latent code $\mathbf{z}$, the time-varying input is parameterized as a distribution $Q(\mathbf{u}_t|\mathbf{x}_t)$, and the decoder network actually receives a sample $\hat{\mathbf{u}}_t$ from this distribution.) These inputs are a critical extension, as autonomous dynamics are likely only a good model for motor areas of the brain, and for specific, tightly-controlled behavioral settings, such as pre-planned reaching movements [19]. Instead, most neural systems and behaviors of interest are expected to reflect both internal dynamics and inputs, to varying degrees. Therefore we focused on the extended model shown in Figure 1(a).

The LFADS objective function is defined as the log likelihood of the data, $\sum_{\mathbf{x}} \log P(\mathbf{x}_{1:T})$, marginalized over all latent variables, which is optimized in VAE setting by maximizing a variational lower bound, $\mathcal{L}$, on the marginal data log-likelihood,

$$\log P(\mathbf{x}_{1:T}) \geq \mathcal{L} = \mathcal{L}^x - \mathcal{L}^{KL}$$

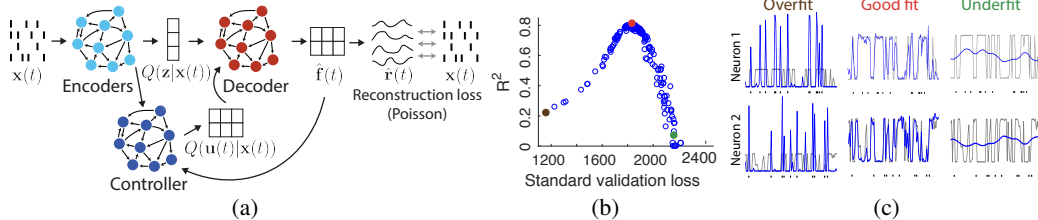

Figure 1: (a) The LFADS architecture for modeling input-driven dynamical systems. (b) Performance of LFADS in inferring firing rates from a synthetic RNN dataset for 200 models with randomly selected hyperparameters. (c) Ground truth (black) and inferred (blue) firing rates for two neurons from three example LFADS models corresponding to points in *b*. Actual spike times are indicated by black dots underneath the firing rate traces. Each plot shows 1 sec of data.

$\mathcal{L}^x$ is the log-likelihood of the reconstruction of the data, given the inferred firing rates $\hat{\mathbf{r}}_t$, and $\mathcal{L}^{KL}$ is a non-negative penalty that restricts the approximate posterior distributions from deviating too far from the (uninformative) prior distribution defined as

$$\mathcal{L}^x = \left\langle \sum_{t=1}^{T} \log\Big(\text{Poisson}(\mathbf{x}_t|\hat{\mathbf{r}}_t)\Big) \right\rangle_{\mathbf{z},\mathbf{u}}$$

$$\mathcal{L}^{KL} = \left\langle D_{KL}\Big(\mathcal{N}\left(\mathbf{g}_0 \mid \boldsymbol{\mu}^{\mathbf{g}_0}, \boldsymbol{\sigma}^{\mathbf{g}_0}\right) \parallel P^{\mathbf{g}_0}\left(\mathbf{g}_0\right)\Big) \right\rangle_{\mathbf{z}} +$$

$$\left\langle D_{KL}\Big(\mathcal{N}\left(\mathbf{u}_1 \mid \boldsymbol{\mu}^{\mathbf{u}}_1, \boldsymbol{\sigma}^{\mathbf{u}}_1\right) \parallel P^{\mathbf{u}_1}\left(\mathbf{u}_1\right)\Big) \right\rangle_{\mathbf{z},\mathbf{u}_1} +$$

$$\left\langle \sum_{t=2}^{T} D_{KL}\Big(\mathcal{N}\left(\mathbf{u}_t \mid \boldsymbol{\mu}^{\mathbf{u}}_t, \boldsymbol{\sigma}^{\mathbf{u}}_t\right) \parallel P^{\mathbf{u}}\left(\mathbf{u}_t|\mathbf{u}_{t-1}\right)\Big) \right\rangle_{\mathbf{z},\mathbf{u}},$$

Here, $\mathbf{g}_0$ represents the initial conditions (ICs) produced by the IC encoder RNN, $\boldsymbol{\mu}^{\mathbf{g}_0}$ and $\boldsymbol{\sigma}^{\mathbf{g}_0}$ are the mean and variance of the IC prior, respectively, and $P^{\mathbf{u}}(\cdot)$ is an autoregressive (AR) prior over inferred inputs [12] with $\boldsymbol{\mu}^{\mathbf{u}}_t$ and $\boldsymbol{\sigma}^{\mathbf{u}}_t$ as the prior means and variances for time $t$, respectively. A more detailed description of the LFADS model is given in Supplemental Section A.

## 2.2 Overfitting on a synthetic RNN dataset

As the complexity and parameter count of neural network models increase, it might be expected that very large datasets are required to achieve good performance. In the face of this challenge, our aim was to test whether such models could be made to function even in cases of limited training data by applying principled HP optimization. For example, adjusting HPs to regularize the system might prevent overfitting, e.g., by limiting the complexity of the learned dynamics via L2 regularization of the recurrent weights of the RNNs, by limiting the amount of information passed through the probabilistic codes $Q(\mathbf{z}|\mathbf{x}_t)$ and $Q(\mathbf{u}|\mathbf{x}_t)$ by scaling the KL penalty, as in [20], or by applying dropout to the networks [21]. Our aim is to regularize the system so it does not overfit, but to also use the least amount of regularization necessary, so that the model still has the capacity to capture as much structure of the data as possible. We first tested whether the model architecture itself was amenable to HP optimization by performing a random HP search. As we show below, the possibility of overfitting via the identity transformation makes such HP optimization a difficult problem.

To precisely measure the performance of LFADS in inferring firing rates, we needed a spiking dataset for which ground truth neural firing rates were known. Real neural data has no ground truth for direct comparison, as there is no "true" measurable firing rate. Common indirect validation measures, such as the likelihood of held-out neurons or behavioral decoding, are not adequate for precisely detecting overfitting. For example, the likelihood of held-out neurons is often a noisy measure and requires assumptions. Similarly, decoding behavior is only a coarse measure of the model's performance, as only a small fraction of neural activity directly correlates with behavior. In addition, behavioral dynamics are often much slower than neural dynamics, making behavioral measures inadequate for testing whether a model captures fine timescale features of neural activity.

To provide a dataset with known neural firing rates, we created synthetic neural data by using an input-driven RNN as a model of a neural system, following [10], Sections 4.2-3. Details of the system

used here are given in Supplemental Section B. We then tested the effect of varying model HPs on our ability to infer the synthetic neurons' underlying firing rates (Figure 1(b)). We trained 200 separate LFADS models in which the underlying model architecture was constant, but we randomly and independently chose the values of five HPs implemented in the publicly-available LFADS codepack. Two HPs were scalar multiples on the KL penalties applied to $Q(\mathbf{z}|\mathbf{x}_t)$ and $Q(\mathbf{u}_t|\mathbf{x}_t)$, two HPs were L2 penalties on the recurrent weights of the generator and controller RNNs, and the last HP set the dropout probability for the input layer and the output of the RNNs.

As shown in Figure 1(b), varying HPs resulted in models whose performance in inferring firing rates ($R^2$) spanned a wide range. Importantly, however, the measured validation loss did not always correspond to accuracy. Figure 1(c) shows ground truth and inferred firing rates for two artificial neurons with their corresponding spike times, for three models that spanned the range of validation losses. Both underfit and overfit models failed to capture the dynamics underlying the neurons' firing rates. Underfit models exhibited overly smooth inferred firing rates, resulting in poor $R^2$ values and reconstruction loss. In contrast, overfit models showed a different failure mode. Rather than modeling the actual structure underlying the firing rates, the networks simply learned to pass spike times through the input channel $Q(\mathbf{u}|\mathbf{x})$, resulting in excellent reconstruction loss for the original, noisy data, but extremely poor inference of the underlying firing rates. Conceptually, the network learned a solution akin to applying an identity transformation to the spiking data. We suspect that this failure mode is more acute with spiking activity, where binned spike counts might be precisely reconstructed by nonlinear transformation of a low-dimensional, time-varying signal. It is worth mentioning that the AR prior mentioned earlier might lessen, but cannot eliminate, this failure mode (all the results in Figure 1(b) are with AR prior applied). The key problem is that the AR prior has a width parameter (described in Supplement Section A.4), which is learnable. Minimizing this width allows the model to get better predictions by overfitting to spikes via inferred inputs. Forcing a minimum AR prior width might prevent overfitting, but might also prevent the model from capturing rapid changes.

Importantly, this failure mode could not be detected through the standard method of validation, which is to hold out entire observations (trials), because those held-out trials are still shown to the network during the inference step and can be used in an identity transformation to achieve accurate reconstruction. Without a reliable validation metric, it is difficult to perform a broad HP search, because it is unclear how one should select amongst the trained models. Ideally one would use performance in inferring firing rates ($R^2$) as a selection metric, but of course ground truth "firing rates" are unavailable for real datasets. One might expect that this failure mode (overfitting via the input channel $Q(\mathbf{u}|\mathbf{x})$) could be sidestepped simply by limiting the capacity of the input channel, either via regularization or by limiting its dimensionality. However, the appropriate dimensionality of the input pathway may be heavily dependent on the dataset being tested (e.g., it may be different for different brain areas), and the susceptibility to overfitting may vary with dataset size, complexity of the underlying dynamics, and the number of neurons recorded. Without knowing *a priori* how to constrain the model, we need the ability to try models with larger capacity and either detect when they overfit, or prevent them from overfitting altogether.

**Denoising autoencoders:** We tested whether existing regularization methods for autoencoders could prevent the overfitting problem described above. For this purpose, we applied two common denoising autoencoder (dAE) approaches [22]: 'Zero masking' (Figure 2(a)), and 'Salt and pepper noise' (Figure 2(b)). We repeated the same experiment presented in Figure 1(b), using both approaches and different input noise levels. Noise level is a critical free parameter, and it is not possible to know the optimal value *a priori*. As shown, depending on the noise level, dAEs could still show pathological

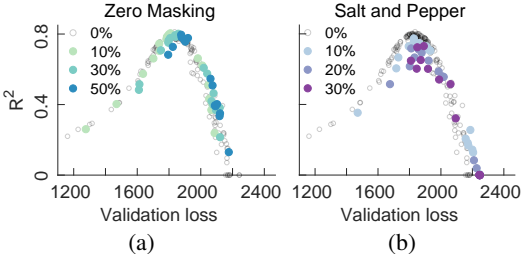

Figure 2: Performance of denoising autoencoders with (a) Zero masking and (b) Salt and pepper noise.

overfitting, which again made standard validation cost an unreliable metric to assess model performance. Furthermore, it can be seen that the higher values of input noise reduced peak performance. Therefore, the remainder of this paper centers on searching for more generalizable solutions that avoid these limitations.

# 3 Validation and regularization methods to prevent overfitting

We developed two complementary approaches to counteract the failure mode of overfitting through identity transformations: 1) a different validation metric to detect overfitting ("sample validation"), and 2) a novel regularization strategy to force networks to model only structure that is shared between dimensions of the observed data ("coordinated dropout").

## 3.1 Sample validation

Our goal with sample validation (SV) was to develop a metric that detects when the networks simply pass data from input to output (e.g., via an identity transformation) rather than modeling underlying structure. Therefore, rather than the standard approach of holding out entire observations of $\mathbf{x}_t$ to compute validation loss [10, 12], SV holds out individual samples randomly drawn from the [*Neurons* × *Time* × *Trials*] data matrix (Figure 3(a)). This approach, sometimes called a "speckled holdout pattern", is a recognized method for cross-validating principal components analysis [23] and has recently

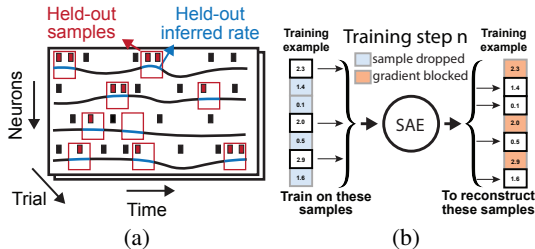

Figure 3: (a) Illustration of sample validation (SV). (b) Illustration of coordinated dropout (CD) for a single training example.

been applied to dimensionality reduction of neural data [24]. We modified our network training in two ways to integrate SV: first, at the network's input, we dropout the held-out samples, i.e., we replace the them by zeros, and linearly scale $\mathbf{x}_t$ by a corresponding amount to compensate for the average decrease in input to the network (similar to [21]). Second, at the network's output, we prevented weight updating using held-out samples (or erroneous updating using the zeroed samples) by blocking backpropagation of the gradient at the specified samples. This prevents the held-out samples (or lack of samples) from inappropriately affecting network training. Finally, because the network still infers rates at the timepoints corresponding to the held-out samples, they can be used to calculate a measure of cross-validated reconstruction loss at a sample-by-sample level. The SV metric consisted of the reconstruction loss averaged over all held-out samples.

## 3.2 Coordinated dropout

The second strategy to avoid overfitting via the identity transformation, coordinated dropout (CD; Figure 3(b)), is based on the reasonable assumption that the observed activity is from a lower dimensional subspace. CD controls the flow of information through the network during training, in order to force the network to model only structure that is shared across input dimensions. At each training step, a random mask is applied to dropout samples of the input. The complement of that mask is applied at the network's output to choose which samples should have their gradients blocked. Thus, for each training step, any data sample that is fed in as input is not used to compute the quality of the network's output. This simple strategy ensures the network cannot learn an identity transformation because individual data samples are never used for self reconstruction. To demonstrate the effectiveness of CD in preventing overfitting, we applied it to the simple case of uncovering latent structure from low-dimensional, noise-corrupted data using a linear autoencoding network (see Supplemental Section C).

## 3.3 Application to the synthetic RNN dataset

Our next aim was to test whether SV and CD could be used to help select LFADS models that had high performance in inferring the underlying spike rates (i.e., did not overfit to spikes), or to prevent the LFADS model from overfitting in the first place. We used the synthetic data described in Section 2.2, and again ran random HP searches (similar to Figure 1(b)). However, in this case, we applied either SV or CD to the LFADS models while leaving other HPs unchanged.

In the first experiment, we tested whether SV provided a more reliable metric of model performance than standard validation loss. When applying SV to the LFADS model, we held out 20% of samples from network training, as described in Section 3.1. Figure 4 shows the performance of 200 models in

inferring firing rates ($R^2$) against the *sample validation loss*, i.e., the average reconstruction loss over the held-out samples. With SV in place, we observed a clear correspondence between the SV loss and $R^2$, which was in sharp contrast to the results when standard validation loss was used to evaluate LFADS models (Figure 1(b)). Models with lower SV loss generally had higher $R^2$, which establishes SV loss as a candidate validation metric for performing model selection in HP searches.

For the second experiment, we tested whether CD could prevent LFADS models from overfitting. In this test we set the "keep ratio" to 0.7, i.e., at each training step the network only sees 70% of the input samples, and uses the complementary 30% of samples to perform backpropagation. Figure 4 shows performance with respect to the *standard validation loss* for the 200 models we trained with CD. Strikingly, we can see a clear correspondence between the standard validation loss and performance, indicating that CD has successfully prevented overfitting during model training. Therefore, the standard validation loss becomes a reliable performance metric when models are trained with CD, and can be used to perform HP search.

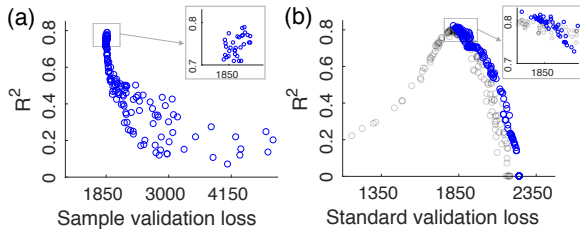

While SV and CD both sidestepped the overfitting problem, we found that models trained with CD had better correspondence between validation loss and performance. With SV, the best models had some variability in the relationship between SV loss and performance ($R^2$; Figure 4(a), inset). With CD, the best models had a more direct correspondence between standard validation loss and performance (Figure 4(b), inset). Because CD produced a more reliable performance measure, we used it to train and evaluate models in the remainder of this manuscript.

Figure 4: (a) Performance of 200 models with the same configuration as in Figure 1(b) plotted against *sample validation loss*. (b) Performance for 200 models plotted against standard validation loss, with CD used during training (blue) or CD off during training (grey, reproduced from Figure 1(b)).

Note that while CD performed well in this test, there may be cases where it is advantageous to use SV in addition, or instead. CD acts as a strong regularizer to prevent overfitting, but it may also result in underfitting. By limiting data to only being seen as either input or output, but never both simultaneously, CD might prevent the model from learning all the structure present in the data. This may have a prominent effect when the number of observation dimensions (e.g., number of neurons) is small relative to the true dimensionality of the latent space. In those cases of limited information, not seeing all the observed data may severely limit the system's ability to uncover latent structure. We believe SV remains a useful validation metric, and future work will test whether SV is a good alternative HP search strategy when the dimensionality of the observed data is more limited.

## 4    Large-scale HP search on neural population spiking activity

Our results with synthetic data demonstrated that SV and CD are effective in detecting and preventing overfitting over a broad range of HPs. Next we aimed to apply these methods to real data to test whether large-scale HP search yields improvements over efforts that use fixed HPs, especially in the case of limited dataset sizes. It is important to note that, although datasets sizes vary substantially between experiments, neuroscientific datasets are often small (e.g., a few hundreds of samples/trials) in comparison to datasets in other fields where deep learning is applied (e.g. thousands or millions of samples). In this section we first describe the experimental data, and then lay out the details of the comparison between fixed HP models and the HP-optimized models. Finally, we present the results of the performance comparison.

### 4.1    Experimental data and evaluation framework

To characterize the effect of training dataset size on the performance of LFADS, we tested LFADS on two real neural datasets. The first dataset is the "Monkey J Maze" dataset used previously [12]. This dataset is considered exceptionally large (2296 trials in total), which allowed us to subsample the data and test the effect of dataset size on model performance. In this data, spiking activity was simultaneously recorded from 202 neurons in primary motor (M1) and dorsal premotor (PMd)

cortices of a macaque monkey as it made 2-dimensional reaching movements with both curved and straight trajectories. A variable delay period allowed the monkey to prepare the movements before executing them. All trials were aligned by the time point at which movement was detectable (movement onset), and we analyzed the time period spanning 250 ms before and 450 ms after the movement onset. The spike trains were placed into 2 ms bins, resulting in 350 timepoints for each trial (2296 trials in total). We then randomly selected 150 neurons from the original 202 neurons recorded, and used those neurons for the entire subsequent analysis.

The second dataset we analyzed is publicly available (`indy_20160426_01` [25]). In this dataset, a monkey made continuous, self-paced reaches to targets in a grid without any gaps or delays. Neural population activity from 181 (sorted) neurons was recorded from M1. There were a total of 715 trials, and we used the same bin size, trial length, and alignment to movement onset as described above for Monkey J Maze.

To evaluate model performance as a function of training dataset size, for each dataset, we trained models using 5%, 10%, 20%, and 100% of the full trial count of 2296. For all sizes below the full trial count, we generated seven separate datasets by randomly sampling from the full dataset (except for the 368 trial count point in the RTT data, which we sampled 3 times). In all cases, 80% of trials were used for model training, while 20% were held-out for validation. We then quantified the performance of each model by estimating the monkey's hand velocities from the model's output (i.e., inferred firing rates). Velocity was decoded using optimal linear estimation [26] with 5-fold cross validation. We used $R^2$ between the actual and estimated hand velocities as the metric of performance.

For each dataset we trained LFADS models in two scenarios: 1) when HPs are manually selected and fixed, and 2) when HP optimization is used.

## 4.2  LFADS trained with fixed HPs

When evaluating model performance as a function of dataset size, it is unclear how to select HPs *a priori*. In our previous work [12] we selected HPs using hand-tuning. We began our performance characterization of fixed HP models using these previously-selected HPs ([12] Supp. Table 1, "Monkey J Maze"). However, we quickly found that performance collapsed for small dataset sizes. Though we did not fully characterize the reason for this failure, we suspect it occurred because the LFADS model previously applied to the Maze data did not attempt to infer inputs (i.e., it only modeled autonomous dynamics), and models without inputs are empirically more difficult to train. The difficulty in training may arise because the sequential autoencoder with no inputs must backpropagate the gradient through two RNNs that are each unrolled for the number of timesteps in the sequence (a very long path). In contrast, models that infer inputs can make progress in learning without backpropagating all the way through both unrolled RNNs, due to the input pathway. Regardless of the reason for this failure, we chose to compare performance for models with inputs, in order to give the fixed-HP models a better shot and avoid a trivial result. Therefore, we switched to a second set of HPs from the previous work, which were used to train models with inputs on a separate task ([12] Supp. Table 1, "Monkey J CursorJump"). These HPs were previously applied to data recorded from the same monkey and the same brain region, but in a different task in which the monkey's arm was perturbed during reaches. We found that the "CursorJump" HPs maintained high performance on the full dataset size and also achieved better performance on smaller datasets than the hand-selected HPs for the Maze data. (Note that while this choice of HPs is also somewhat arbitrary, it illustrates the lack of principled method for choosing fixed HPs when applying LFADS to different datasets.)

## 4.3  LFADS trained with HP optimization

To perform HP optimization, we integrated a recently-developed framework for distributed optimization, Population Based Training (PBT) [27]. Briefly, PBT is a method to train many models in parallel, and it uses evolutionary algorithms to select amongst the set of models and optimize HPs. PBT was shown to consistently outperform methods like random HP search on several neural network architectures, while requiring the same level of computational resources as random search [27]. Thus it seemed a more efficient framework for large-scale hp optimization than random HP search. We implemented the PBT framework based on [27] and integrated it with LFADS to perform HP

optimization with a few tens of models on a local cluster. We applied CD while training LFADS models, and used the standard validation loss as the performance metric for model selection in PBT.

Two classes of model HPs might be adjusted: HPs that set the network architecture (e.g., number of units in each RNN), and HPs that control regularization and other training parameters. In our HP optimization, We fixed the network architecture HPs to match the values used in the fixed-HPs scenario, and allowed the other HPs to vary. Specifically, we allowed PBT to optimize learning rate, keep ratio (i.e., fraction of the data samples that are passed into the model using CD) and five different regularizers: L2 penalties on the generator (L2 Gen scale) and controller (L2 Con scale) RNNs, scaling factors for KL penalties applied to $Q(\mathbf{z}|\mathbf{x}_t)$ (KL IC scale) and $Q(\mathbf{u}_t|\mathbf{x}_t)$ (KL CO scale), and dropout probability (Table 1).

PBT enables different schedules for different HPs during training, and this results in better performance than random HP searches [27]. We generally observed that the learning rate and KL penalty scales began at the higher end of their ranges (Table 1) and decreased over the course of training. Conversely, L2 and keep ratio often increased, and dropout often remained low throughout training.

Table 1: List of HPs searched with PBT

| HP | Value/Range | Initialization |
|---|---|---|
| L2 Gen scale | $(5, 5e4)$ | log-uniform |
| L2 Con scale | $(5, 5e4)$ | log-uniform |
| KL IC scale | $(0.05, 5)$ | log-uniform |
| KL CO scale | $(0.05, 5)$ | log-uniform |
| Dropout | $(0, 0.7)$ | uniform |
| Keep ratio | $(0.3, 0.99)$ | 0.5 |
| Learning rate | $(10^{-5}, 0.02)$ | 0.01 |

## 4.4 Results

For LFADS models trained with fixed HPs, we found that performance significantly worsened as smaller datasets were used for training (Figure 5(a),(d), red). This was expected and illustrates previous concerns regarding applying deep learning methods with limited datasets [8]. For the HP-optimized models (Figure 5(a),(d), black), performance with the largest dataset is comparable to the fixed-HPs model, which confirms that HP optimization is not critical when enough data is available. However, as the amount of the training data decreased, models with optimized HPs maintained high performance even up to an impressive 10-fold reduction in training data size. (It is important to note that the Monkey J Maze and RTT datasets were chosen because they are larger than typical neuroscientific datasets, which often fall in the range where HP optimization appears to be critical.)

We also quantified the average percentage performance improvement achieved by optimizing HPs, relative to fixed HPs (Figure 5(b),(e)). As the training data size decreased, HP search became critical in improving performance. To better illustrate this improvement, we compared the decoded position trajectories from a fixed-HP model trained using 10% of the data (184 trials; Figure 5(c), *top*) against trajectories decoded from a HP-optimized model (Figure 5(c), *bottom*). The HP-optimized model led to significantly better estimation of reach trajectories.

These results demonstrate that with effective HP search, deep learning models can still maintain high performance in inferring neural population dynamics even when limited data is available for training. This greatly extends the applicability of LFADS to scenarios previously thought to be challenging due to the limited availability of data for model training.

Beyond the application to spiking neural data demonstrated in this paper, the proposed techniques should be generally applicable to over-complete/sparse autoencoder architectures [28] or when forecasting time series from sparse data, especially when HP or architecture searches are important.

## 5 Conclusion

We demonstrated that a special case of overfitting occurs when applying SAEs to spiking neural data, which cannot be detected through using standard validation metrics. This lack of a reliable validation metric prevented effective HP search in SAEs, and we demonstrated two solutions: sample validation (SV) and coordinated dropout (CD). As shown, SV can be used as an alternate validation metric that, unlike standard validation loss, is not susceptible to overfitting. CD is an effective regularization technique that prevents SAEs from overfitting to spiking data during training, which allows even the standard validation loss to be effective in evaluating model performance. We illustrated the effectiveness of SV and CD on synthetic datasets, and showed that CD leads to better correlation between the validation loss and the actual model performance. We then demonstrated the challenge

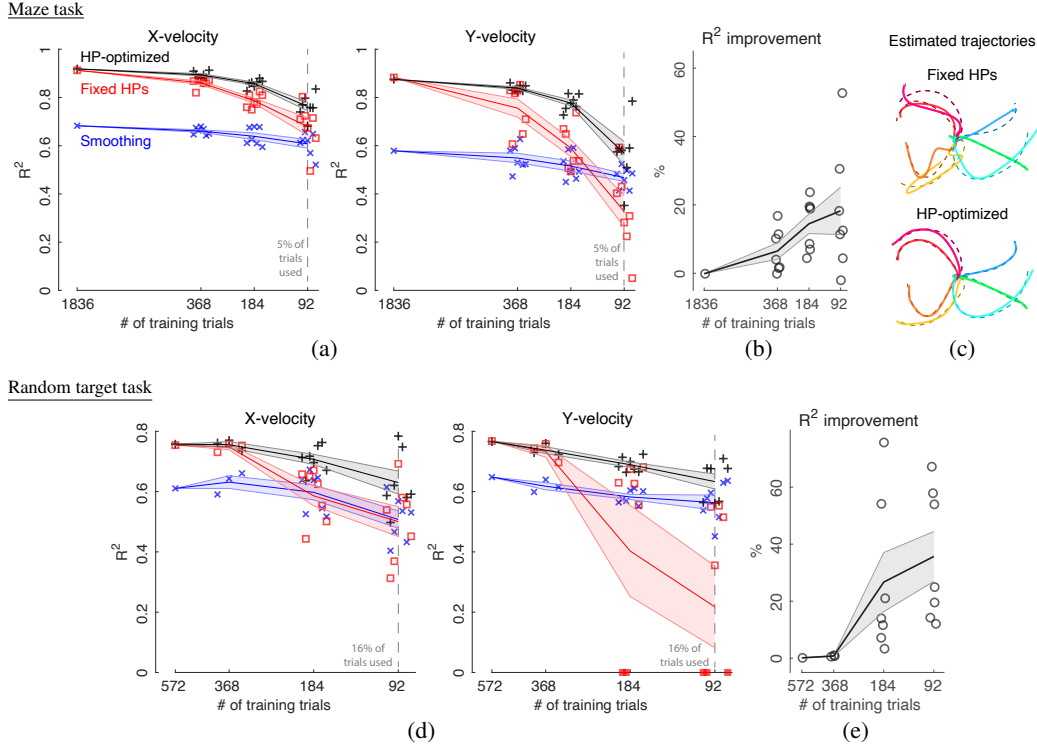

Figure 5: (a) Performance in decoding hand velocity after smoothing spikes with a Gaussian kernel (blue, $\sigma = 60$ ms standard deviation), applying LFADS with fixed HPs (red), and applying LFADS with HP optimization (black). Note that the dataset size is decreasing from left to right. Left and right panels show decoding for hand X- and Y-velocities, respectively. Lines and shading denote mean $\pm$ standard error across multiple models for the same dataset size (random draws of trials, note we only have one sample for the full dataset). (b) Percentage improvement in performance from HP-optimized models, relative to fixed HPs, averaged across all the models for each dataset size. (c), *top* Examples of estimated (solid) and actual (dashed) reach trajectories for LFADS with fixed HPs (the model with median performance on 184 trials). (c), *bottom* Reach trajectories when HP optimization was used. Trajectories were calculated by integrating the decoded hand velocities over time. (d), (e) Same results for the random target task (RTT) dataset.

of achieving good performance with the LFADS model when the training dataset size is small. With CD in place, effective HP search can greatly improve the performance of LFADS. Most importantly, we demonstrated that with effective HP search we could train SAE models that maintain high performance, even up to an impressive 10-fold reduction in the training data size. Applications of SV and CD are not limited to the LFADS model, but should also be useful for other autoencoder structures when applied to sparse, multidimensional data.

## Acknowledgements

We thank Chris Rozell, Ali Farshchian, Raghav Tandon, Andrew Sedler, and Lahiru Wimalasena for their comments on the paper. This work has been supported by NSF NCS 1835364, and DARPA Intelligent Neural Interfaces program.

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
