[Supplementary Material]

# A Full LFADS model

The following description is adapted from [1].

LFADS is based on a variational autoencoder (VAE) [2], adapted to model sequences (as in [3]). The VAE consists of two main components, an encoder and a generator (aka. decoder). The encoder takes as input the binned spike data, $\mathbf{x}$, and transforms them into a conditional distribution over $\mathbf{z}$, $Q(\mathbf{z}|\mathbf{x})$, where $\mathbf{z}$ is a vector of stochastic latent variables, and $Q(\mathbf{z}|\mathbf{x})$ is a learnable approximation of the posterior distribution of the generator, $Q(\mathbf{z}|\mathbf{x}) \approx P(\mathbf{z}|\mathbf{x}) = P(\mathbf{x}|\mathbf{z})P(\mathbf{z})/P(\mathbf{x})$. The generator denoted by $P(\mathbf{x}|\mathbf{z})$, then receives samples drawn from the posterior distribution $Q(\mathbf{z}|\mathbf{x})$, and maps them back into the approximation of the data $\hat{\mathbf{x}}$. Over the course of training, the autoencoder tries produce $\hat{\mathbf{x}}$ that resemble $\mathbf{x}$.

The network's objective function is to maximize the likelihood of the data while minimizing the Kullback-Leibler (KL) divergence between the encoding distribution $Q(\mathbf{z}|\mathbf{x})$ and an uninformative Gaussian prior $P(\mathbf{z})$, over all data points.

For the rest of this section, we denote an affine transformation ($\mathbf{v} = \mathbf{W}\,\mathbf{u} + \mathbf{b}$) from a vector $\mathbf{u}$ to a vector $\mathbf{v}$ as $\mathbf{v} = \mathbf{W}(\mathbf{u})$, we use $[\cdot,\cdot]$ to denote vector concatenation, and we denote a temporal update of a recurrent neural network receiving an input as $\text{state}_t = \text{RNN}^a(\text{state}_{t-1}, \text{input}_t)$, for an RNN named 'a', where different superscripts denote different RNN networks which do not share parameters. Gated Recurrent Units (GRU) [4] were used to implement all the RNN networks.

## A.1 LFADS Encoder

We denote the spiking neural data for $T$ trials by $\mathbf{x}_{1:T}$. We assume that $\mathbf{x}_{1:T}$ are observed samples from a Poisson process with underlying rates $\mathbf{r}_{1:T}$. Through the autoencoding process, LFADS infers a set of low-dimensional latent "factors" $\mathbf{f}_{1:T}$. The factors are produced by the generator RNN, as described in section A.2. The firing rates can be constructed through a affine transformation of the factors, followed by an exponential nonlinearity, $\mathbf{r}_{1:T} = \exp(\mathbf{W}^{\text{rate}}(\mathbf{f}_{1:T}))$.

The binned spiking data $\mathbf{x}_{1:T}$ is fed into the bidirectional RNN encoder network $\text{RNN}^{g0}$, with final cell state $\mathbf{E}^{g0}$. The output of this network is used to set the initial state of the generator RNN (described in A.2). Specifically, the initial state of the generator RNN is defined by the approximate posterior $Q^{g_0}(\mathbf{g}_0|\mathbf{x})$, with means and diagonal covariance matrices taken as an affine transformation of the (concatentated) final states of the bidirectional RNNs $\mathbf{E}^{g0}$.

$$\boldsymbol{\mu}^{g_0} = \mathbf{W}^{\mu^{g_0}}(\mathbf{E}^{g0}) \tag{1}$$

$$\boldsymbol{\sigma}^{g_0} = \exp\left(\frac{1}{2}\mathbf{W}^{\sigma^{g_0}}(\mathbf{E}^{g0})\right). \tag{2}$$

The initial conditions $\hat{\mathbf{g}}_0$ are then sampled from the resulting distribution

$$\hat{\mathbf{g}}_0 \sim Q^{g_0}(\mathbf{g}_0|\mathbf{x}) = \mathcal{N}(\mathbf{g}_0 \mid \boldsymbol{\mu}^{g_0}, \boldsymbol{\sigma}^{g_0}) \tag{3}$$

To allow LFADS to model data generated by a potentially non-autonomous underlying process, a set of time-varying "inferred inputs" $\mathbf{u}_{1:T}$ are also provided as the input to the generator RNN. This expands the latent variables to $\mathbf{z} = \{\mathbf{g}_0, \mathbf{u}_{1:T}\}$. Thus the approximate posterior distribution for LFADS consists of two conditional Gaussian distributions, one for $\mathbf{g}_0$ and one for $\mathbf{u}_t$.

The approximate posterior distribution for the inferred inputs $\mathbf{u}_t$ is obtained through a second bidirectional encoder RNN network ($\text{RNN}^{ce}$) with time-varying cell state $\mathbf{E}_t^{\text{con}}$, and a unidirectional controller RNN ($\text{RNN}^{\text{con}}$). $\text{RNN}^{ce}$ processes the binned spiking data, and its time-varying cell state $\mathbf{E}_t^{\text{con}}$ is fed to the controller RNN along with a delayed version of the latent factors, $\mathbf{f}_{t-1}$.

$$\mathbf{c}_t = \text{RNN}^{\text{con}}(\mathbf{c}_{t-1}, [\mathbf{E}_t^{\text{con}}, \mathbf{f}_{t-1}]). \tag{4}$$

The initial state of the controller network, $\mathbf{c}_0$, is defined as a trainable bias initialized to the 0 vector.

The inferred inputs seen by the generator, $\hat{\mathbf{u}}_t$, are samples from diagonal Gaussian distributions where mean and log-variance are given by an affine transformation of the cell state of the controller RNN, $\mathbf{c}_t$,

$$\hat{\mathbf{u}}_t \sim Q^u(\mathbf{u}_t|\mathbf{x}) = \mathcal{N}(\mathbf{u}_t|\boldsymbol{\mu}_t^u, \boldsymbol{\sigma}_t^u) \tag{5}$$

with

$$\boldsymbol{\mu}_t^{\mathrm{u}} = \mathbf{W}^{\mu^{\mathrm{u}}}(\mathbf{c}_t) \tag{6}$$

$$\boldsymbol{\sigma}_t^{\mathrm{u}} = \exp\left(\frac{1}{2}\mathbf{W}^{\sigma^{\mathrm{u}}}(\mathbf{c}_t)\right). \tag{7}$$

An information bottleneck is imposed on the controller output to the generator by limiting the dimensionality of $\mathbf{u}_t$ (a hyperparameter), and by applying a KL penalty described in Section A.3. Here, we used diagonal Gaussian priors for $\mathbf{g}_0$ and $\mathbf{u}_1$, and an autoregressive Gaussian prior (described in Section A.4) for $\mathbf{u}_t$ with $t > 1$.

## A.2  LFADS Generator

An RNN network (RNN$^{\mathrm{gen}}$) is used to implement the generator, while the factors are obtained through affine transformation of the generator states (denoted by $\mathbf{g}_t$), $\mathbf{f}_{1:T} = \mathbf{W}^{\mathrm{fac}}(\mathbf{g}_{1:T})$. The generator's initial state $\mathbf{g}_0$ is obtained by drawing samples from the learnable posterior distribution $Q^{\mathbf{g}_0}(\mathbf{g}_0|\mathbf{x})$.

The LFADS generator, including the inferred inputs, is described by the following equations. The initial condition for the generator is sampled from its approximate posterior according to (3). The inferred inputs, $\hat{\mathbf{u}}_t$, are sampled from the approximate posterior according to (5) and fed into the network for each time step $t = 1, \ldots, T$, and the generator states are updated from the old states and the current inferred inputs,

$$\mathbf{g}_t = \mathrm{RNN}^{\mathrm{gen}}\left(\mathbf{g}_{t-1}, \hat{\mathbf{u}}_t\right) \tag{8}$$

$$\mathbf{f}_t = \mathbf{W}^{\mathrm{fac}}(\mathbf{g}_t) \tag{9}$$

$$\mathbf{r}_t = \exp\left(\mathbf{W}^{\mathrm{rate}}\left(\mathbf{f}_t\right)\right) \tag{10}$$

$$\tag{11}$$

## A.3  The loss function

The loss function has been described in the main text. We describe it here again for completeness. The loss function is defined as the log likelihood of the data, $\sum_{\mathbf{x}} \log P(\mathbf{x}_{1:T})$, marginalized over all latent variables, which is optimized in a VAE setting by maximizing a variational lower bound, $\mathcal{L}$, on the marginal data log-likelihood,

$$\log P(\mathbf{x}_{1:T}) \geq \mathcal{L} = \mathcal{L}^x - \mathcal{L}^{KL}, \tag{12}$$

where

$$\mathcal{L}^x = \left\langle \sum_{t=1}^{T} \log\left(\mathrm{Poisson}(\mathbf{x}_t|\mathbf{r}_t)\right) \right\rangle_{\mathbf{g}_0, \mathbf{u}_{1:T}} \tag{13}$$

$$\mathcal{L}^{KL} = \left\langle D_{KL}\left(\mathcal{N}\left(\mathbf{g}_0 \mid \boldsymbol{\mu}^{\mathbf{g}_0}, \boldsymbol{\sigma}^{\mathbf{g}_0}\right) \parallel P^{\mathbf{g}_0}\left(\mathbf{g}_0\right)\right) \right\rangle_{\mathbf{g}_0} +$$

$$\left\langle D_{KL}\left(\mathcal{N}\left(\mathbf{u}_1 \mid \boldsymbol{\mu}_1^{\mathrm{u}}, \boldsymbol{\sigma}_1^{\mathrm{u}}\right) \parallel P^{\mathbf{u}_1}\left(\mathbf{u}_1\right)\right) \right\rangle_{\mathbf{g}_0, \mathbf{u}_1} +$$

$$\left\langle \sum_{t=2}^{T} D_{KL}\left(\mathcal{N}\left(\mathbf{u}_t \mid \boldsymbol{\mu}_t^{\mathrm{u}}, \boldsymbol{\sigma}_t^{\mathrm{u}}\right) \parallel P^{\mathrm{u}}\left(\mathbf{u}_t|\mathbf{u}_{t-1}\right)\right) \right\rangle_{\mathbf{g}_0, \mathbf{u}_{1:T}}. \tag{14}$$

The brackets denote marginalizations over the sub-scripted variables.

## A.4  Autogressive prior for inferred inputs

A zero-mean autoregressive (AR) process with one time lag is defined by

$$s(t) = \alpha s(t-1) + \epsilon_s(t), \tag{15}$$

with $0 \leq \alpha < 1$ and noise variable $\epsilon_s(t)$ drawn from $\mathcal{N}(0, \sigma_\epsilon^2)$. This prior is used for inferred inputs $\mathbf{u}_t$ with $t > 1$, separately for each dimension, where the AR process autocorrelations and variances are initialized to user-defined values and trained along with rest of the parameters in LFADS.

## B  Generation of synthetic data using an input-driven RNN

To precisely measure the performance of LFADS in inferring firing rates, we needed a spiking dataset that reflects a dynamical system where ground truth neural firing rates were known. Therefore we created synthetic neural data by using an input-driven RNN as a model of a neural system, following [5], Sections 4.2-3. Briefly, we simulated an RNN with $N = 50$ artificial units whose temporal evolution followed

$$\tau \, \dot{\mathbf{y}}(t) = -\mathbf{y}(t) + \gamma \, \mathbf{W}^{\mathrm{y}} \tanh(\mathbf{y}(t)) + \mathbf{B} \, \mathbf{q}(t), \tag{16}$$

with $\mathbf{y}(t)$ being an $N = 50$-length vector, $\gamma = 2.5$, and $\tau = 0.025s$. $\mathbf{W}^{\mathrm{y}}$ specifies the RNN's recurrent connectivity, with elements drawn from $\mathcal{N}(0, 1/N)$. $\mathbf{q}(t)$ was a two-dimensional time-varying input to the system, with samples at each point drawn independently from $\mathcal{N}(0, 1)$. Elements of $\mathbf{B}$ were also drawn independently from $\mathcal{N}(0, 1)$. Spiking data were then generated by drawing Poisson samples from the rates of the RNN's artificial units (i.e., $\tanh(\mathbf{y}(t))$) after shifting and scaling these rates to span the range 0-30 spikes/sec. We simulated 4000 trials, 1 sec each, which included 400 unique "conditions". For a given condition, individual trials began with the RNN in the same initial state, but used different random inputs $\mathbf{q}$ and different random draws of Poisson spiking. for the tests based on synthetic data, we binned spiking data into 10 ms bins, resulting in trials that were 100 steps long, and split the data into 3200 training and 800 validation trials.

## C  Testing coordinated dropout using a linear autoencoder

To illustrate the effectiveness of coordinated dropout (CD), we applied CD to a simple example, the case of attempting to uncover latent structure from low-dimensional, noise-corrupted data using a linear autoencoding network. To generate synthetic data with low-D structure, we created a $D$-dimensional vector of factors $\mathbf{f}(t)$ by sampling from $\mathcal{N}(0, 1)$. We then projected the factors onto an $M$-dimensional ($D < M$) vector $\mathbf{y}_{true}(t)$ using a readout matrix $\mathbf{W}$, where elements of $\mathbf{W}$ were set by sampling from $\mathcal{N}(0, 1)$. Our observed data, $\mathbf{y}$, was then taken as a noise-corrupted version of $\mathbf{y}_{true}$, where $\mathbf{y} = \mathbf{y}_{true} + \mathcal{N}(0, 1)$.

Our goal was then to recover $\mathbf{y}_{true}$ from $\mathbf{y}$, assuming no knowledge of $\mathbf{y}_{true}$, using a simple linear autoencoder. $\hat{\mathbf{y}} = \hat{\mathbf{U}} \times \mathbf{y}$. In this exercise, we only wanted to demonstrate the autoencoder's behavior when its capacity was far higher than the data dimensionality. Therefore we did not constrain the autoencoder's dimensionality, that is, $\hat{\mathbf{U}}$ was an $[M \times M]$ matrix. In training the autoencoder, the objective was to minimize the reconstruction loss, i.e., $\arg\min_{\hat{\mathbf{U}}} ||\hat{\mathbf{y}} - \mathbf{y}||$. The weights of $\hat{\mathbf{U}}$ were randomly initialized and trained via backpropagation and stochastic gradient descent. While this training approach attempts to minimize the error in reconstructing the observed data (i.e., $loss_{\mathrm{recon}} = ||\hat{\mathbf{y}} - \mathbf{y}||$), an ideal approach would of course minimize the error in reconstructing the unobserved, noise-free data (i.e., $loss_{\mathrm{true}} = ||\hat{\mathbf{y}} - \mathbf{y}_{true}||$).

We set $D = 5$, $M = 40$, and used 1000 and 200 samples for training and for validation, respectively. Figure 1(a) shows the results. As no constraints were applied to the dimensionality of the autoencoder, it unsurprisingly converges to a trivial solution that simply outputs $\mathbf{y}$, i.e.,

Figure 1:  (a) Training loss (red) and true loss (blue) for a simple linear autoencoder applied to noise-corrupted, low-dimensional data. (b) The resulting weight matrix. (c) Loss when CD is used for training. (d) Resulting weight matrix.

setting $\hat{\mathbf{U}}$ to the identity matrix (Figure 1(b)). This solution of course achieves excellent reconstruction loss on noisy data even for held out observations. However, in terms of estimating $\mathbf{y}_{true}$ (blue curve in Figure 1(a)), this solution is heavily overfit.

Figure 1(c) shows the results with the same approach after CD is added. We set the "keep ratio" to 0.8, i.e., on each training step, the network saw 80% of the data samples, and we only allowed the gradient to backpropagate on the complimentary 20% of samples. While the reconstruction loss on the observed data $loss_{\mathrm{recon}}$ (red curve) is much worse than before, the reconstruction loss on

the unobserved data $loss_{\text{true}}$ is dramatically improved, as the system does not overfit over time. As shown in Figure 1(d), in this simple linear autoencoder, CD is effectively equivalent to preventing the diagonal weights in $\hat{\mathbf{U}}$ from being trained. However, because CD is applied at the level of the data input/output, it can be used for arbitrary network architectures, even when the "diagonal" is not clearly defined.