[Reviews · NeurIPS 2019]

Reviewer 1



Authors provide novel approaches to calculate cross-validated reconstruction loss by using one of two proposed solutions: Sample validation and Coordinated dropout described above. The ideas are first described with the help of a synthetically generated dataset and experimental results on Monk This paper would be much stronger if the ideas were demonstrated on multiple real datasets. As well as demonstrating comparable performance using only x% of multiple datasets. In the current organization, the ideas are first demonstrated on synthetically generated data. It is not clear why the "Monkey J Maze" is not used right from the beginning, instead of spending significant portion of the data in describing the synthetic data generation process. Synthetic data is unconvincing especially in an unsupervised learning setting. These look like sound ideas and would be generally applicable for autoencoders in any domain. So if getting neurological data is challenging, data from other domains may be considered. It would be a good idea to use existing regularization methods for autoencoders as baselines e.g., Denoising autoencoders. While the proposed methods look sound, the motivation for using completely new techniques should be explained. While the proposed methods are promising, this paper appears to be research in progress, and would benefit from demonstration of more experimental results. ## Comments post rebuttal and review discussion Having reviewed the rebuttal and other reviews, I think its ok to accept. I am convinced with the need to demonstrate on synthetic data, it helps to explicitly make the case is made for readers unfamiliar with the domain for whom this is not obvious. Having further experimental results on real data helps to make the case as well. Good to see that the experimental results hold on the 2nd "Random Target" dataset. Ideally I would have liked to see experimental results on more than 2 datasets. It was also good to see baselines on dAE on synthetic dataset. Ideally, this should be demonstrated on the 2 real datasets as well. Partial CD (splitting data into input only, shared and output only) is a good suggestion. However, I don't understand why the experiments were conducted on a smaller subset of the data. As this was not part of the original paper, it is okay to ignore and leave it out. If a comparison of Partial CD will be included in the final version of the paper, we would need to see (i) complete results on Partial CD (ii) a clear description of exactly what splits were used for experiments. (iii) a better description of conditions under which partial CD does better than full CD [Fig. 3 in the rebuttal that shows improvement in 8 of 10 models.] (iv) description of when SV is also additionally needed.

Reviewer 2



A method to optimize hyperparameters (HPs) for sequential autoencoders (SAEs) are tested to prevent overfitting to infer structure from high-dimensional spiking data from biological neurons. The authors find that SAEs applied to spiking neural data are prone to a particular form of overfitting that cannot be detected using standard validation metrics, which prevents standard HP searches. Two potential solutions are explored: an alternate validation method (“sample validation”) and a novel dropout regularization method. Effectiveness of these regularizers is demonstrated on monkey motor cortex spike data. The biggest disadvantage of the model seems to be, that the method is tailored to spiking neuron model. It is therefore not clear if the proposed regularizers could be extended to other learning problems. Have the authors explored this possibility? A brief discussion would increase the utility of the paper.

Reviewer 3



The authors present a significant and original contribution to training autoencoder models in neuroscience that may be broadly applicable in machine learning applications. The issue of pathological overfitting and the proposed solutions are clearly described, although inclusion of some additional important details would improve clarity (see below). The training methods described in this paper will be useful to neuroscientists and machine learning practitioners training autoencoder models. The motivating example is a sequential autoencoding model LFADS. In LFADS (previously described elsewhere), the generative model is an RNN that generates low-D factors underlying higher-dimensional Poisson count data. At each time point, random inputs are fed into the RNN. The model is trained using amortized variational inference using an RNN encoding network to approximate the variational posterior over the initial condition of the generator RNN, and a “controller” RNN that approximates the posterior over the input perturbations. Fitting this model to different datasets with different dimensionalities and amounts of training data requires hyperparameter tuning. However, the authors show that when using random hyperparameter search, the “best” hyperparameters chosen via validation loss may correspond to a model that is pathologically overfit, where the controller RNN learns inferred inputs that simply reproduce spikes in the data. The authors propose two training schemes to overcome the pathological overfitting. In the first, called sample validation, random elements of the neuron by time by trial input array are dropped out at both the input to the approximate posterior (encoder network) and the output of the generative model, such that the gradients with respect to the dropped out elements are blocked. This scheme has been applied previously in cross validation for PCA. Next, the authors propose an original training scheme called coordinated dropout. In coordinated dropout, the data is split into two parts. The first split of the data (e.g. 70% of data) is used as input to the encoder network, while the rest of the data is used to compute the training objective and gradients. This method relies on the modeling assumption that the data can be described by low-d factors, and its utility is demonstrated on a linear autoencoder example. On a synthetic example, the authors show that when either method is used in training across random hyperparameters the models no longer exhibit pathological overfitting. Notably, the proposed coordinated dropout method produces a strong correlation between heldout loss and training performance. Accordingly, the validation loss can be used to select the proper hyperparameters. Finally, the authors show the power of using coordinated dropout when fitting to LFADS on real data with hyperparameter search. The models trained with coordinated dropout show improved decoding performance, especially on smaller dimensional datasets. The improved performance of the model on smaller dimensional datasets is notable. While the motivation for the new training methods, methods, and examples are clear, the paper could be improved with some additional details. - Can the authors state the training objective with the hyperparameters as an equation in the manuscript? As written, I think I can piece together the training objective - the ELBO with penalties on the two different KL terms, one for the inferred inputs and one for initial condition, plus penalties on the weights on the controller and generator RNNs. However, including the equation is critical to make this precise and clear to the general reader. - Can the authors also state the validation loss and how it is computed? - Additionally, it would be useful to fully describe the LFADS model, at least in an appendix. This will increase clarity to readers unfamiliar with LFADS. - Is there a prior on the input p(u)? In Pandarinath, Nature Methods 2018 an autoregressive prior was applied on p(u). Does the model still exhibit pathological overfitting with that autoregressive prior included? Comments -What settings of the random hyperparameters provided “good” fits? It would be interesting to include a discussion of this, including how this might vary across dataset size. -Is full-split coordinated dropout necessary, or could you also split the data into input only, shared, and output only splits? --------------------------- After reading the other reviewers and author feedback and conferring with reviewers, I have increased my score by a point. The experiments and details in the author feedback have improved the clarity and significance of the submission. I encourage the authors to include the comparison with dAEs and model details in the final version of the manuscript, and to follow Reviewer 1's guidance about whether or not to include Partial CD in the final version.

[Author Response · NeurIPS 2019]

**We appreciate the thoughtful feedback.** All reviewers noted that our *sample validation* (SV) and *coordinated dropout*
(CD) methods were novel with broad applicability. New analyses, clarifications, and proposed modifications are below.
**R1:** *Paper would be much stronger if ideas*
*were demonstrated on multiple real datasets* Done

**Fig. 1.** **a**) Rand Targ task   **b**) # of trials for 47 experiments [1]

(**Fig. 1a**). We used an open dataset [1] with a Ran-
dom Target task (different lab and experiment). We
found similar results to orig. Fig. 5, including the
range where HP opt helps, and the gap between op-
timized and fixed HPs. **R1:** *Description of typical*
*dataset sizes would help motivate the criticality of*
*the issue*; *Single small dataset is insufficient to estab-*
*lish general efficacy.* Agreed, we'll discuss. Typical
sizes largely vary, so for context we'll show the trial counts for 47 experiments from the open dataset (**Fig. 1b**; [1]).
These dataset sizes are typical, and many are in the range where HP opt is important. Note: our original dataset
(1836 trials) is actually *exceptionally large*, chosen so we could characterize HP opt vs. dataset size. **R1:** *Not clear*
*why "Monkey J Maze" is not used from the beginning... Synthetic data is unconvincing.* This is a key point. It is
important to clarify the necessity of tests on synthetic data, and may also help for readers without neural data experience.
**The synthetic data is critical - without it, it is very challenging to determine whether an approach results in**
**pathological overfitting**. Real neural data has no ground truth for direct comparison - there is no "true", measurable
firing rate. Common validation measures are problematic for detecting overfitting: *1)* Held-out likelihood of observed
data is somewhat noisy and requires assumptions. *2)* Decoding behavior, as we do, is a rough measure: only a small
fraction of neural activity correlates with behavior, and behavioral dynamics are quite slow. A precise characterization of
overfitting (orig. Fig. 1) and of the effectiveness of SV/CD (orig. Fig. 4) would be very challenging with real data. Since
SV & CD are the key innovations, we must thoroughly characterize them using data with a ground truth, and synthetic
data are the best option. To speed manuscript, we will move all synthetic data generation details to a supplement. **R1:**
*Existing regularization like denoising autoencoders (dAEs) should also be used as baselines. Motivation for completely*
*new techniques should be explained.* Great suggestion. We tested dAEs (**Fig. 2**),

**Fig. 2.** Denoising AE results

and motivation is now easily explained in the context of these results. We re-
peated orig. Fig. 1b using two common dAE approaches for discrete data: 'Zero
masking' and 'Salt and pepper noise' [2]. Important points: *1)* dAEs have a free
parameter (noise level). *2)* Depending on its setting, dAEs can still show patho-
logical overfitting. *3)* Some settings can even reduce performance. *4)* It is not
possible to know how to set dAE noise *a priori*. Our methods bypass these limi-
tations (see orig. Fig. 4), providing a reliable metric to measure (SV) or completely block (CD) pathological overfitting.
**R2:** *Discuss if method can be extended to other data sets.* Good point, will add. Techniques should be applicable when
forecasting time series from sparse data, especially when HP or architecture searches are important. Examples are usage
at electrical vehicle charging stations, taxi/rideshare calls, etc.. We're currently trying to apply this to generative models
for LIDAR/RADAR data for autonomous cars (e.g., following [3]). **R3:** *Would raise my score with the inclusion of some*
*details that were missing... complete formulation of the generative model and inference procedure.* Good suggestion. We
will add this information. R3's description of objective was accurate. **R3:** *State validation loss and how it is computed...*
*Useful to fully describe LFADS model, at least in appendix.* Apologies for omissions, will add. **R3:** *Does the model still*
*exhibit pathological overfitting with AR prior included?* Yes, and we were surprised by this (all the results in paper
are with AR prior included). Key problem is AR prior is learnable, and model can adapt it to get better predictions by
overfitting to spikes via inputs. Forcing a minimum AR prior autocorrelation might prevent overfitting, but might also
prevent the model from capturing rapid changes. **R3:** *What HP settings provided "good" fits? Would be interesting to*
*include a discussion, including how this might vary across dataset size.* Agreed, including settings/ranges will be helpful.
Further, these methods enabled dynamic HP opt (changing HPs during training) using population based training [4].
This somewhat surprisingly yields even higher performance by learning schedules for different HPs (e.g., KL penalty
is set high during early training, but decreases over time). We'll add this discussion. **R3:** *Is full-split CD necessary,*
*or could you also split the data into input only, shared, and output only splits?* This is very interesting, we've been

**Fig. 3.** Partial CD

thinking about this also. The proposed 'Partial CD' approach might help when observed number
of neurons is similar to the underlying dimensionality, and fully splitting data via CD may limit
training. Without Full CD, though, a method is needed to detect/prevent overfitting. SV fills this
role. As suggested, we turn CD on, and then allow some fraction of the data (searchable HP) to
be shared as input and output. Preliminary tests on small sets of randomly drawn neurons (Monkey
J Maze data, 25 per draw) show promising results: Partial CD outperforms Full CD in 8/10 models
tested. Thorough tests will help delineate conditions where Partial CD helps.

[1] J E O'Doherty et al. http://doi.org/10.5281/zenodo.583331, 2017.      [2] P Vincent et al. *J. Mach. Learn. Res.*,
11:3371–3408, 2010.      [3] L Caccia et al. *arXiv:1812.01180*.      [4] M Jaderberg et al. *arXiv:1711.09846*, 2017.


[Meta-Review · NeurIPS 2019]

The paper demonstrates that the sequential autoencoder that's becoming popular in neuroscience is prone to overfitting and propose solutions to address this overfitting. It is overall a good paper. A couple of small issues could be corrected to improve the quality: - Baselines such as dAEs that were in the feedback should be put in the main paper along with running baselines on the real datasets - A discussion of alternatives and baselines to motivate the solution